# A transcriptome-wide association study identifies *PALMD* as a susceptibility gene for calcific aortic valve stenosis

Sébastien Thériault[1,2], Nathalie Gaudreault [1], Maxime Lamontagne[1], Mickael Rosa[1], Marie-Chloé Boulanger[1], David Messika-Zeitoun[3,4], Marie-Annick Clavel [1], Romain Capoulade[1], François Dagenais[1], Philippe Pibarot[1], Patrick Mathieu[1] & Yohan Bossé [1,5]

Calcific aortic valve stenosis (CAVS) is a common and life-threatening heart disease and the current treatment options cannot stop or delay its progression. A GWAS on 1009 cases and 1017 ethnically matched controls was combined with a large-scale eQTL mapping study of human aortic valve tissues ($n = 233$) to identify susceptibility genes for CAVS. Replication was performed in the UK Biobank, including 1391 cases and 352,195 controls. A transcriptome-wide association study (TWAS) reveals *PALMD* (palmdelphin) as significantly associated with CAVS. The CAVS risk alleles and increasing disease severity are both associated with decreased mRNA expression levels of *PALMD* in valve tissues. The top variant identified shows a similar effect and strong association with CAVS ($P = 1.53 \times 10^{-10}$) in UK Biobank. The identification of *PALMD* as a susceptibility gene for CAVS provides insights into the genetic nature of this disease, opens avenues to investigate its etiology and to develop much-needed therapeutic options.

[1] Institut universitaire de cardiologie et de pneumologie de Québec—Université Laval, Quebec City, QC G1V 4G5, Canada. [2] Department of Molecular Biology, Medical Biochemistry and Pathology, Laval University, Quebec City, QC G1V 0A6, Canada. [3] Cardiology Department, AP-HP, Bichat Hospital, 75018 Paris, France. [4] INSERM U698 and University Paris 7, 75018 Paris, France. [5] Department of Molecular Medicine, Laval University, Quebec City, QC G1V 0A6, Canada. Correspondence and requests for materials should be addressed to Y.B. (email: yohan.bosse@criucpq.ulaval.ca)

Calcific aortic valve stenosis (CAVS) is the most common valvular heart disease (2% in > 65-year-old individuals)[1]. It is characterized by a progressive remodeling and calcification of the aortic valve, leading to stenosis and heart failure. There is a long period of disease progression before CAVS becomes severe and symptomatic, which provides a window of opportunity for intervention[2]. Unfortunately, conventional cardiovascular drugs, such as statins and angiotensin-converting-enzyme inhibitors, are unable to stop or delay the progression of CAVS[3–6]. If severe CAVS is left untreated, the survival of affected patients is dramatically shortened following the onset of symptoms[7]. The only effective treatments available for symptomatic patients with severe CAVS are invasive and costly surgical or transcatheter interventions[8]. Finding new molecular targets to halt or slow disease progression is an urgent priority[9].

Our molecular and genetic understanding of CAVS is currently limited. A strong genetic component is suggested by early-onset forms of the disease, familial aggregation, and genetic epidemiology studies[10–13]. A number of susceptibility genes were identified[14–18]. Genome-wide gene expression studies of normal and stenotic human valves have highlighted biological processes that are involved in CAVS[19,20]. However, full integration of genome-wide association study (GWAS) results with a large-scale transcriptomic data set in aortic valves has not been performed.

In this study, we combine GWAS and valve expression quantitative trait loci (eQTL) results to identify the molecular drivers of CAVS. The GWAS includes 1009 severe cases of tricuspid CAVS confirmed at surgery and 1017 ethnically matched controls with normal aortic valve that underwent cardiac surgery (QUEBEC-CAVS cohort). In parallel, we perform a large-scale eQTL mapping study on human aortic valve tissues ($n = 233$), which allows us to undertake a transcriptome-wide association study (TWAS)[21] on CAVS. The TWAS identifies *PALMD* on chromosome 1p21.2 as a candidate causal gene, a finding that we further corroborate by Mendelian randomization and colocalization analyses. We then validate the genetic association on 1p21.2 in UK Biobank and confirm that the discovery consists of a non-coronary artery disease (CAD) gene that is specific to CAVS. This is an important step forward to understand the molecular mechanisms underpinning the heritable risk of CAVS.

## Results

**GWAS in the QUEBEC-CAVS cohort**. Table 1 shows the clinical characteristics of 1009 cases and 1017 controls that passed quality controls (QCs). Mean and standard deviation for age was 71.7 ± 8.3 years and 64% were men. About half (53%) of CAVS cases and the majority of controls (98%) had CAD. The GWAS evaluated 7,732,680 single-nucleotide polymorphisms (SNPs) after imputation. The quantile-quantile plot indicated no inflation of observed test statistics (Supplementary Fig. 1). GWAS results are illustrated in Fig. 1a. No SNP reached genome-wide significance ($P_{GWAS} = 5 \times 10^{-8}$). Eight loci had at least one SNP with $P_{GWAS} < 1 \times 10^{-6}$ (Supplementary Table 1 and Supplementary Fig. 2).

**Valve eQTL mapping study**. An eQTL mapping study was then conducted in the most disease-relevant tissue to study CAVS, i.e., human aortic valve. Valve eQTLs were calculated in 233 patients that passed QC for genotyping and gene expression. A total of 10,598 independent valve eQTLs were identified at $P_{eQTL} < 1 \times 10^{-8}$ (Supplementary Data 1). Overall, these eQTLs involved 2277 genes/probes significantly associated with one to 41 independent SNPs ($r^2 < 0.8$) (Supplementary Fig. 3a). On average, a single SNP explained 26.5% of the gene/probe expression variance. However, 29.1% of the eQTL-SNPs explained more than 30% of the expression variance (Supplementary Fig. 3b).

**Table 1 Clinical characteristics of the QUEBEC-CAVS cohort**

| Characteristics | Cases (n = 1009) | Controls (n = 1017) |
|---|---|---|
| Age (years) | 72.5 ± 8.4 | 70.9 ± 8.1 |
| Gender (% male) | 63.6 | 64.7 |
| Aortic valve area (cm$^2$) | 0.77 ± 0.30 (19) | NA |
| Mean gradient (mm Hg) | 41.1 ± 16.0 (64) | NA |
| Concomitant coronary artery bypass grafting (%) | 53.3 | 98.3 |
| Diabetes (%) | 32.4 | 32.4 |
| Hypertension (%) | 75.6 | 75.0 |
| BMI (kg/m$^2$) | 28.6 ± 5.4 | 27.7 ± 4.7 |
| Cholesterol (mmol/L) | 4.2 ± 1.1 (30) | 3.9 ± 1.0 (23) |
| Triglycerides (mmol/L) | 1.5 ± 0.9 (32) | 1.5 ± 0.8 (24) |
| LDL-C (mmol/L) | 2.2 ± 0.9 (36) | 2.0 ± 0.8 (31) |
| HDL-C (mmol/L) | 1.3 ± 0.4 (32) | 1.1 ± 0.3 (28) |
| Lipid-lowering treatment (%) | 72.6 | 88.0 |
| eGFR (mL/min/1.73 m$^2$) | 66.9 ± 18.2 (1) | 68.6 ± 16.7 (3) |

Continuous variables are mean ± SD. Number of missing values is shown in parentheses when applicable. eGFR was estimated using the CKD-EPI equation[50]

**Identification of *PALMD* by TWAS**. A TWAS[21] was then performed using the GWAS and valve eQTL data sets. A Manhattan plot showing transcriptome-wide association in valve tissue with CAVS is shown in Fig. 1b. Only one probe-CAVS association corresponding to the *PALMD* gene reached genome-wide significance ($P_{TWAS} = 0.00007$). *PALMD* is located on chromosome 1p21.2 and the top GWAS SNPs were the same as the top valve eQTL-SNPs (Fig. 2a). rs6702619, a genotyped variant, was the SNP in this locus most significantly associated with both CAVS (odds ratio (OR) = 1.29, 95% confidence interval (CI) 1.14–1.46, $P_{GWAS} = 6.12 \times 10^{-5}$) and the expression of *PALMD* ($P_{eQTL} = 5.82 \times 10^{-33}$). In addition, formal Bayesian colocalization[22] revealed a posterior probability of shared signals (PP4) of 0.96, which confirms that the GWAS and valve eQTL signals share the same variants at the *PALMD* locus. rs6702619 is located 65.2 kb from the transcriptional start site of *PALMD*. The valve eQTL rs6702619-*PALMD* indicated that the risk allele for CAVS ("G") is associated with lower mRNA expression levels of *PALMD* in valve tissues (Fig. 2b), suggesting that lower expression increases the risk of CAVS. Variants located within 1 Mb of *PALMD* that increased its expression tended to decrease the risk of CAVS (Fig. 2c). Concordantly, protein expression of PALMD in aortic valves was lowered in homozygotes GG compared to homozygotes TT for rs6702619 (Supplementary Fig. 4). rs6702619 is a common SNP (minor allele frequency of 47.7% in Europeans from the 1000 Genomes Project) and the estimated population-attributable risk indicates that more than 12.5% of cases in our population could be attributed to this common variant.

**Genetically lowered *PALMD* expression is associated with CAVS**. A Mendelian randomization analysis suggested a causal role of lower *PALMD* expression on CAVS risk, without evidence of pleiotropy. Variants located within 200 kb of *PALMD* and associated with *PALMD* gene expression were selected using stepwise regression (Supplementary Table 2) to create an instrument for Mendelian randomization analyses. Effect on *PALMD* gene expression was inversely associated with the effect on CAVS risk without evidence of pleiotropy ($P = 0.0036$; Egger intercept $P = 0.25$; Fig. 3a). The direction of the association was the same when only the lead variant (rs6702619) was included ($P = 1.37 \times 10^{-4}$). Considering the number of tests performed, none of the SNPs in the instrument was significantly associated with the expression of the three nearest genes (*PLPPR4*, *FRRS1*,

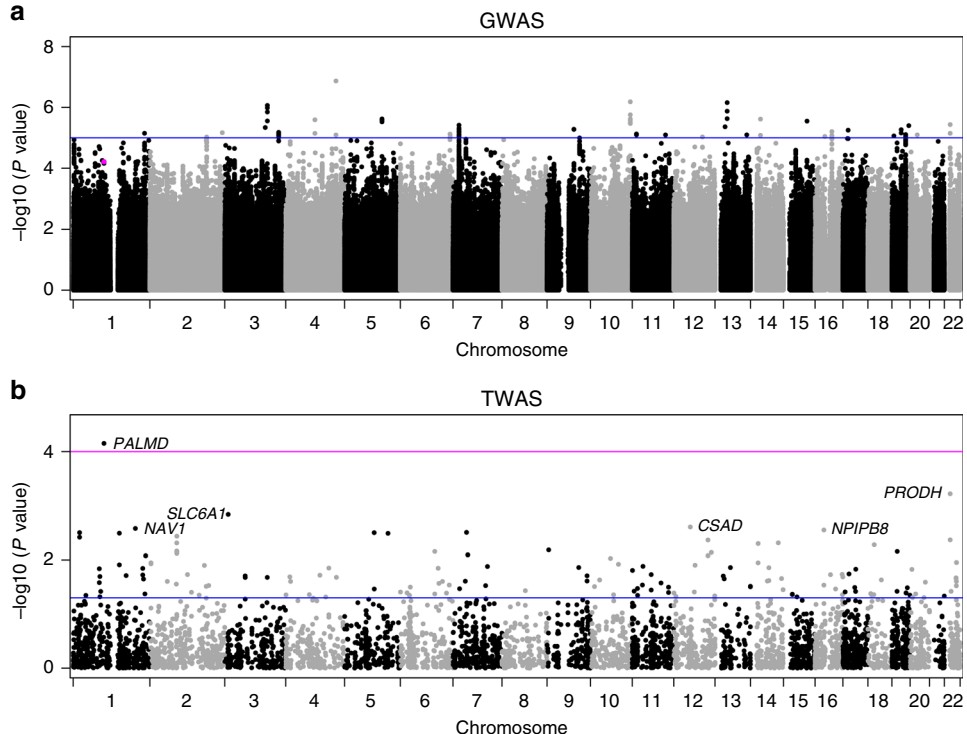

**Fig. 1** Manhattan plots showing the GWAS in QUEBEC-CAVS and TWAS results. **a** Genetic associations with CAVS observed in 1009 cases and 1017 controls. The y-axis represents P value in −log10 scale. The horizontal blue line indicates a P value of $1 \times 10^{-5}$. The magenta dot indicates rs6702619. **b** Transcriptome-wide association in valve tissue with CAVS. P values for gene expression-CAVS associations are on the y-axis in −log10 scale. The blue horizontal line represents $P_{TWAS}$ of 0.05. The magenta horizontal line represents the genome-wide significant threshold used in this study ($P_{TWAS} <$ 0.0001). Annotations for the top significant probes are indicated

and *AGL*; P > 0.0042 or 0.05/12). Together, these results indicated that risk variants on 1p21.2 confer susceptibility to CAVS through downregulation of *PALMD* in aortic valve tissues. The relationship between *PALMD* expression levels and CAVS severity was also consistent with this direction of effect. Lower normalized, age- and sex-adjusted *PALMD* expression was associated with smaller aortic valve area (P = 0.0027), higher mean transvalvular gradient (P = 0.0001), and higher peak transvalvular gradient (P = $8.13 \times 10^{-5}$; Fig. 4).

**PALMD eQTL in Genotype-Tissue Expression**. We additionally looked for *PALMD* eQTL in the Genotype-Tissue Expression (GTEx) data set[23]. rs6702619-*PALMD* association was not significant in 44 tissues from GTEx, suggesting that the change in expression is specific to aortic valve tissue. Evaluating other SNPs within or near *PALMD* revealed significant eQTL with this gene in the pancreas, tibial nerve, and subcutaneous adipose tissue. However, these eQTL-SNPs were not in linkage disequilibrium (LD) with CAVS-associated variants.

**Replication in UK Biobank**. We then replicated our newly associated locus using data recently released by the UK Biobank, which included 1391 CAVS cases and 352,195 unaffected individuals all of European ancestry. We observed a very strong association with an almost identical effect on CAVS risk for our top SNP rs6702619 with an OR of 1.27 (95% CI 1.18–1.37; Fig. 5a). The association was also replicated for rs7543130, a genotyped variant in perfect LD with rs6702619. At the genome-wide scale level, only two loci were genome-wide significant in UK Biobank (Fig. 5b): *LPA* on 6q25.3-q26 previously associated with aortic valve calcification[17]; and *PALMD* on 1p21.2 identified in this study. Mendelian randomization using the effect on CAVS

risk as estimated in the UK Biobank also pinpoints *PALMD* expression as a causal factor (P = $1.18 \times 10^{-5}$; Egger intercept P = 0.10; Fig. 3b). In UK Biobank, the population-attributable risk for rs6702619 was estimated at 12.3%.

## Discussion

CAVS is a common and life-threatening heart disease with no drug that can stop or delay its progression. Elucidating the genetic factors underpinning CAVS is an urgent priority to find new therapeutic targets[9]. Major landmarks in genetics of CAVS include the discoveries of *NOTCH1*[24] and *LPA*[17]. However, genetic variants in these genes accounted for a small number of cases and a low population-attributable risk. Here we mapped a new susceptibility locus for CAVS on chromosome 1p21.2 and identified *PALMD* (palmdelphin) as the candidate causal gene. *PALMD* was revealed using a TWAS[21], which combined a GWAS of 1009 cases and 1017 ethnically matched controls with the first large-scale QTL mapping study on human aortic valve tissues (n = 233). The CAVS risk alleles and increasing disease severity were both associated with lowered mRNA expression levels of *PALMD* in valve tissues. The top variant explained up to 12.5% of the population-attributable risk and showed similar effect and strong association with CAVS (P = $1.53 \times 10^{-10}$) in UK Biobank comparing 1391 cases and 352,195 controls. The identification of *PALMD* as a susceptibility gene for CAVS provides new insights about the genetic nature of this disease and opens new avenues to investigate its etiology and develop much-needed therapeutic options.

The top GWAS SNP in this study (rs6702619) was recently identified by the EchoGen consortium to be associated with aortic root diameter in a large GWAS meta-analysis, including up to 46,533 individuals[25]. In our study, rs6702619 was not associated with aortic root diameter (P = 0.18), which was available in a

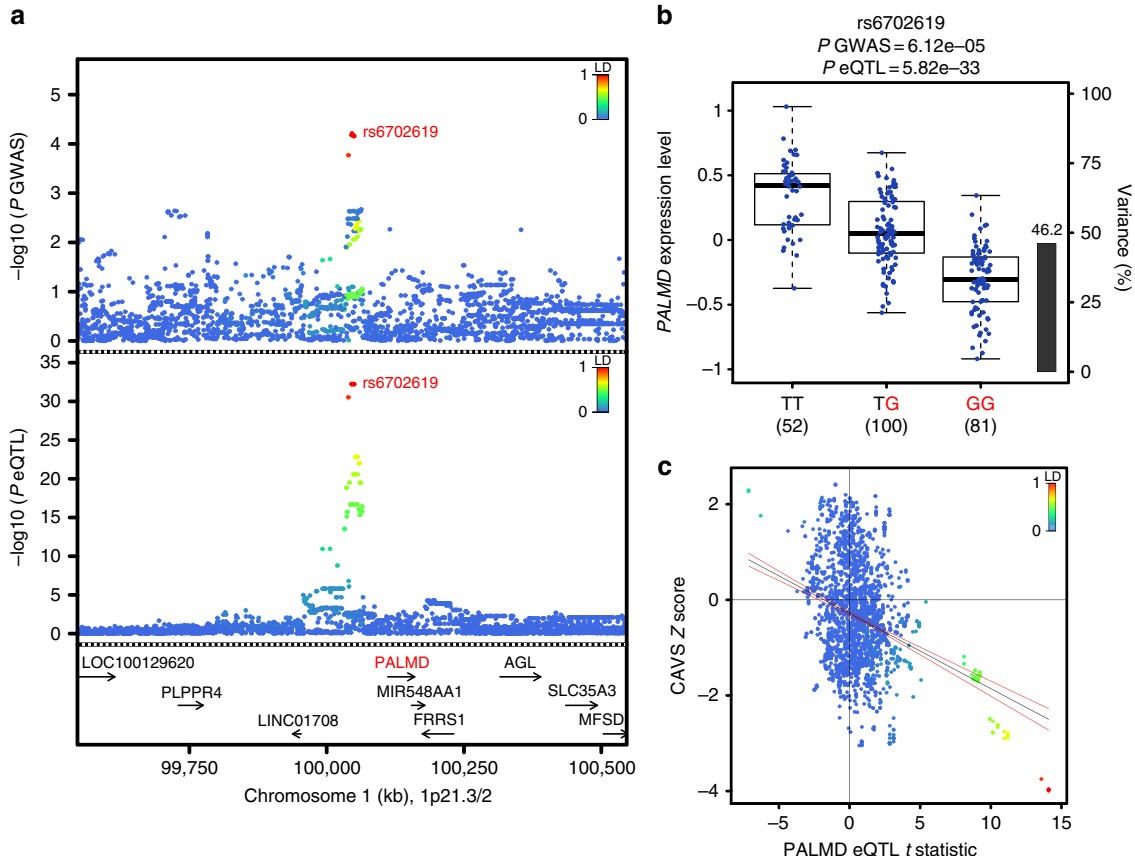

**Fig. 2** *PALMD* is the candidate causal gene on the 1p21.2 CAVS susceptibility locus. **a** GWAS and valve eQTL results surrounding *PALMD* on chromosome 1p21. The upper panel shows the genetic associations with CAVS. The bottom panel shows the valve eQTL statistics for the *PALMD* gene. The extent of linkage disequilibrium (LD; $r^2$ values) for all SNPs with rs6702619 is indicated by colors. The location of genes in this locus is illustrated at the bottom. **b** *PALMD* gene expression levels in aortic valves by genotyping groups for SNP rs6702619. The *y*-axis shows *PALMD* mRNA expression levels. The *x*-axis denotes the three genotype groups. The number of individuals is indicated in parentheses. The CAVS risk allele is illustrated in red. Boxplot boundaries represent the first and third quartiles, whiskers are the most extreme data point, which is no more than 1.5 times the interquartile range, and the center mark represents the median. The black bar and the right *y*-axis indicate the variance in *PALMD* gene expression explained by rs6702619. **c** Scatterplot of the 1p21.2 susceptibility locus showing SNP associations with CAVS and *PALMD* gene expression in aortic valve tissues. The *y*-axis represents variant association with CAVS (*Z* score). The *x*-axis shows association with *PALMD* gene expression (*t* statistic). Variants are colored based on the degree of LD ($r^2$) with the top CAVS-associated variant rs6702619. The blue line is the regression slope with 95% confidence interval (red lines)

subgroup of 900 participants (698 cases and 202 controls). The EchoGen consortium also identified a SNP in *CACNA1C* (calcium channel, voltage-dependent, ʟ type, alpha 1C subunit) associated with aortic root diameter[25], which we recently identified as a susceptibility gene of CAVS[18]. As indicated in the later study, rs6702619 is located within enhancer histone marks and DNase-hypersensitive sites in Encyclopedia of DNA Elements (ENCODE) data[26]. However, rs6702619 was not found to act as *cis*-eQTL in whole blood, monocytes, and myocardial tissue[25]. In this study, we found *PALMD* consistently expressed in human aortic valve tissues and rs6702619 strongly associated with mRNA expression levels of *PALMD*, providing a potential functional mechanism of how genetic variants on 1p21.2 may increase CAVS susceptibility. Furthermore, a Mendelian randomization analysis suggested that *PALMD* expression is causally associated with the risk of CAVS.

The frequency of rs6702619 is high in our French Canadian population (minor allele frequency (MAF) = 47%) and the modest effect size (OR = 1.29) is consistent with those observed for other complex traits. The estimated population-attributable risk indicates that more than 12.5% of cases in our population and 12.3% of cases in UK Biobank could be attributed to this common variant. Of note, the frequency of rs6702619 varies

greatly between ethnic groups (Supplementary Fig. 5). In the 1000 Genomes Project, allele "G" had a frequency of 48% in Europeans, 8% in Africans, 7% in East Asians, and 25% in South Asians. This may explain some of the discrepancy in risk of CAVS among ethnic groups[27,28].

*PALMD* is a distant homolog of a small paralemmin protein family[29]. *PALMD* is expressed in many tissues, but most abundantly in cardiac and skeletal muscle[30]. It is a mainly cytosolic protein, localized predominantly in actin filaments, and may be implicated in plasma membrane dynamics and cell shape control as other members of this family[29,31]. However, its molecular and cellular functions remain largely unknown. *PALMD* was recently shown to promote myoblast differentiation and muscle regeneration[32]. Considering the common embryologic origin of the aortic valve and the aortic root, which both arise from the secondary heart field, a potential role on smooth muscle and valve interstitial cell differentiation could explain the association with the two phenotypes[33,34]. *PALMD* was also identified as a pro-apoptotic gene induced by p53 in response to DNA damage in osteosarcoma cell lines[35]. However, its role in cell death was not confirmed in other cell types[32].

The controls for the QUEBEC-CAVS GWAS were patients with CAD without valvulopathy. This strategy limited our ability

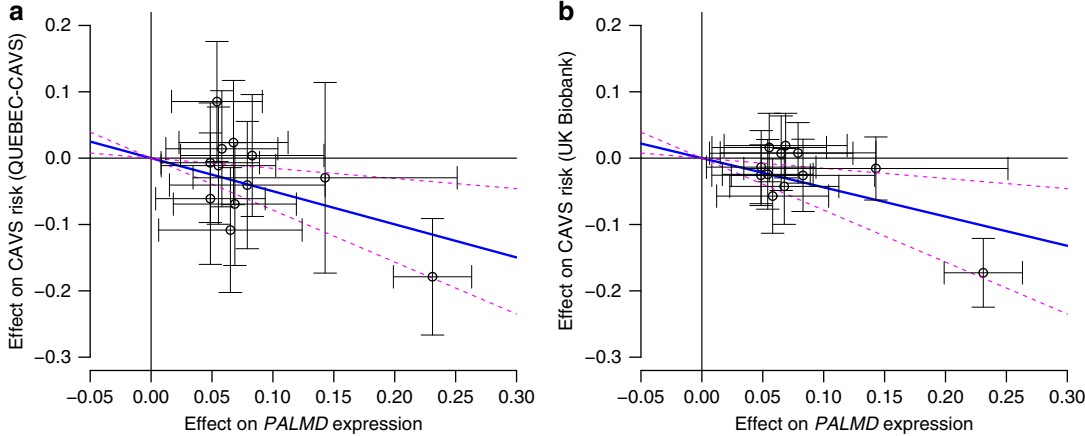

**Fig. 3** Mendelian randomization analysis of the association between *PALMD* gene expression and CAVS risk. Each circle represents 1 of 12 SNPs located within 200 kb of *PALMD* selected for association with *PALMD* gene expression ($P < 0.05$) using stepwise regression. The blue line is the regression slope using the Wald method. The magenta dashed lines represent 95% confidence intervals from bootstrap. **a** Effect on CAVS risk from QUEBEC-CAVS cohort against effect on *PALMD* gene expression ($P = 0.0036$). **b** Effect on CAVS risk from UK Biobank against effect on *PALMD* gene expression ($P = 1.18 \times 10^{-5}$)

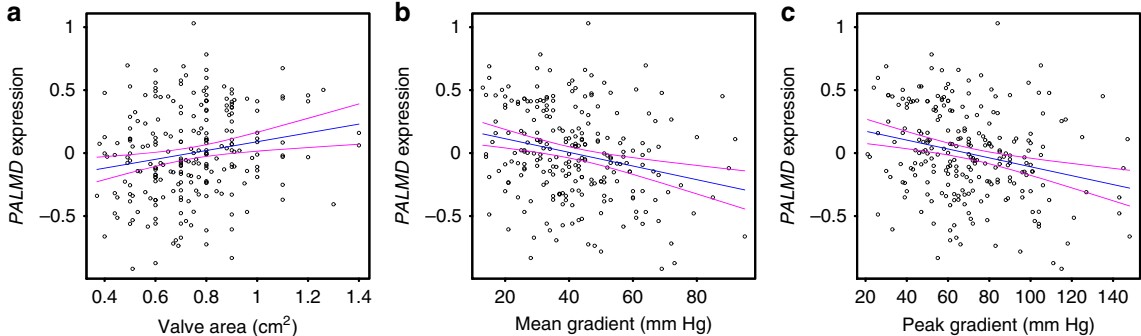

**Fig. 4** Relationship between *PALMD* expression levels and CAVS severity. *PALMD*-normalized, age- and sex-adjusted mRNA expression levels in 239 aortic valve tissues according to CAVS disease severity assessed by **a** aortic valve area ($P = 0.0027$), **b** mean ($P = 0.0001$), and **c** peak transvalvular gradients ($P = 8.13 \times 10^{-5}$). The blue lines represent the slopes obtained by linear regression with 95% confidence interval (magenta lines)

to detect shared genetic variants with CAD. On the other hand, genes that are specific for CAVS are more likely to be found. This is illustrated by comparing the *LPA* locus (a known locus associated with both CAD and CAVS) and the *PALMD* locus in the UK Biobank with and without adjustment for CAD. The association with CAVS was not influenced at the *PALMD* locus with an OR = 1.29 ($P = 1.02 \times 10^{-10}$) and OR = 1.27 ($P = 6.29 \times 10^{-11}$) with and without adjustment for CAD, respectively. In contrast, the strength of association at the *LPA* locus was lowered after adjustment for CAD (OR = 1.39, $P = 3.54 \times 10^{-8}$ vs. OR = 1.56, $P = 2.58 \times 10^{-14}$). The variant identified at the *PALMD* locus is therefore expected to act via pathways not involved in the risk of CAD. rs6702619 was indeed not associated with CAD in a recent large GWAS meta-analysis ($P = 0.96$)[36]. In the QUEBEC-CAVS cohort, the association of rs6702619 with CAVS stratified by the presence of CAD in cases showed consistent results with overlapping effect sizes. In addition, the association of the variant with the presence of CAD in CAVS cases was not significant ($P = 0.15$). Of note, rs6702619 was not associated with aortic valve calcification measured by computed tomographic scanning in a large GWAS, including 6942 individuals of European ancestry ($P = 0.557$)[17]. This suggests that calcification is not the main mechanism by which *PALMD* expression modulates CAVS progression and could rather appear only in later stages of the disease.

Our study has some limitations. First, only individuals of European ancestry were included in the analysis, therefore results cannot be generalized to other ethnic groups. Second, the power in our GWAS analysis was limited by the number of available samples. However, expression data on the most disease-relevant tissue and replication in a large population-based cohort confirmed the robustness of the findings. Third, aortic valve tissue was only available in stenotic valves; patterns of expression of *PALMD* in normal valves might differ.

In conclusion, using a TWAS approach with expression data in aortic valve tissue, we identified a new CAVS susceptibility gene, *PALMD*, with replication in a large prospective population-based cohort. Further analyses of gene expression in valve tissue and disease severity suggested that the identified variant acts by decreasing the expression of *PALMD*, a finding supported by Mendelian randomization. Further studies are warranted to elucidate the exact mechanism of action and evaluate the potential for targeted therapeutic interventions.

## Methods

**Study cohort.** Blood samples and aortic valves were collected from patients with severe aortic valve stenosis undergoing aortic valve replacement at the *Institut universitaire de cardiologie et de pneumologie de Québec* (QUEBEC-CAVS). Only cases with tricuspid nonrheumatic CAVS were included. No severe regurgitation or other severe valvular heart diseases were present. In parallel, an ethnically matched control group was recruited from patients that underwent cardiac surgery, mostly for isolated coronary artery bypass (> 98%). Other indications for surgery in the control group included heart transplant, tumor removal, aortic endoprosthesis, and interatrial communication. Absence of CAVS was confirmed by echocardiography. This cohort of control patients was also matched in a 1:1 ratio with cases for age,

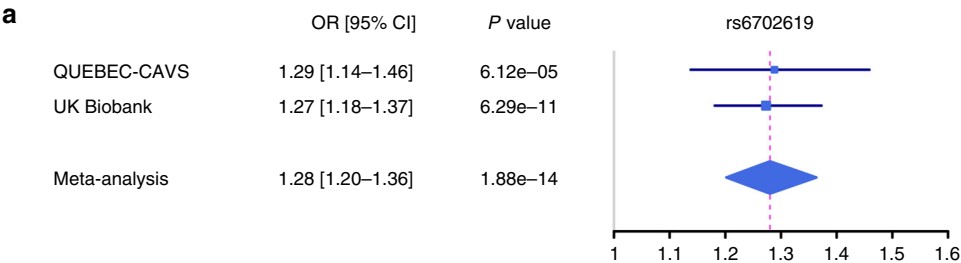

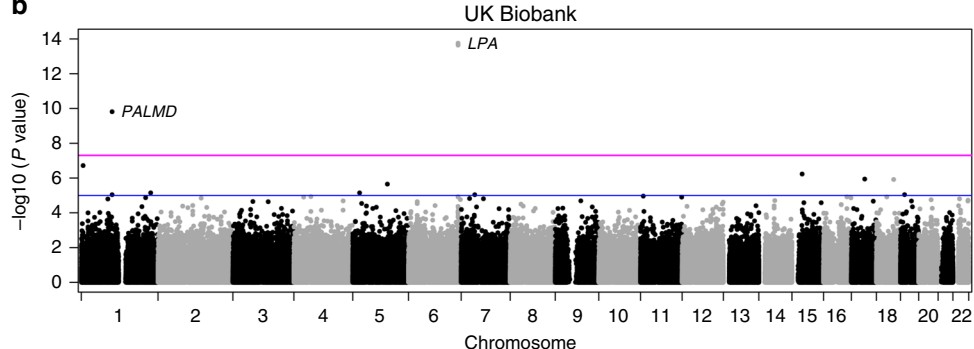

**Fig. 5** Replication in UK Biobank. **a** Forest plot of overall effect size for rs6702619 in the discovery (QUEBEC-CAVS) and validation (UK Biobank) cohorts. The blue filled squares represent the odds ratio (OR) for each cohort with 95% confidence intervals illustrated by horizontal lines. The gray vertical line represents an OR of 1.0 and the dashed magenta line is the OR of the meta-analysis. **b** Manhattan plot showing the GWAS in UK Biobank comparing 1391 CAVS cases and 352,195 controls. The horizontal blue and magenta lines indicate $P_{GWAS}$ values of $1 \times 10^{-5}$ and $5 \times 10^{-8}$, respectively. Three SNPs were significant at the $P_{GWAS}$ threshold of $5 \times 10^{-8}$: rs74617384 and rs10455872 at the *LPA* locus and rs7543130 (in perfect LD with rs6702619) at the *PALMD* locus

gender, type 2 diabetes, and hypertension. Patients with a history of severe valvular heart disease (at any of the four valves), with significant aortic valve regurgitation (grade > 2) or with end-stage renal disease (estimated glomerular filtration rate < 15 mL/min/1.73 m²) were excluded. CAVS patients and controls were free of congenital heart defects (including bicuspid aortic valve), except for seven participants with interatrial communication (three cases and four controls). All patients signed an informed consent for the realization of genetic studies. The study was approved by the ethics committee of the *Institut universitaire de cardiologie et de pneumologie de Québec*. Demographics, anthropometric measurements, lifestyle factors, previous and current medical history, current medication, and blood pressure measurements were collected. Having a history of myocardial infarction, coronary artery stenosis on coronary angiography, or documented myocardial ischemia was defined as CAD. In addition, fasting plasma lipids and creatinine were measured. CAVS cases underwent a comprehensive Doppler echocardiographic examination. The transvalvular gradient was calculated using the modified Bernoulli equation and the aortic valve area calculated with the continuity equation. For this study, 1033 cases and 1037 controls were available.

**Genome-wide association study**. Blood samples were collected and DNA was extracted from frozen buffy coat. Whole-genome genotyping was performed using the Illumina HumanOmniExpress BeadChip. Standard genotyping QCs were performed. SNP genotyping data were filtered for call rate < 97%, low-quality loci with 10th percentile of Illumina GenCall score ≤ 0.1, Hardy–Weinberg equilibrium $P < 1 \times 10^{-7}$, MAF < 1%, and different call rate between cases and controls ($P < 1 \times 10^{-6}$). A total of 613,862 SNPs passed QC checks. Samples were excluded after consideration for the 10th percentile of Illumina GenCall score ≤ 0.2, genotype completion rate < 95%, outlier heterozygosity rate (F > 0.20), genotypic and phenotypic gender mismatch, genetic background outliers detected by principal component analysis with HapMap subjects as population reference panel, and unexpected duplicates and genetic relatedness (first-degree relatives) evaluated by identity-by-state using PLINK[37]. After the QC filters, 1009 cases and 1017 controls were available for subsequent analyses. Supplementary Tables 3 and 4 provide a summary of genotyping QCs on SNPs and samples. Genotypes were then imputed with the Michigan Imputation Server[38] using the Haplotype Reference Consortium version 1 (HRC.r1-1) data[39] as reference set (2016-03-31). Variants with an $r^2$ value of ≤ 0.3 or MAF < 1% were removed from further analysis. A total of 7,732,680 imputed SNPs passed QC. Genetic association tests were performed using additive logistic regression models based on expected genotype counts (dosages) as implemented in the software SNPTEST v2.5.2 [40], adjusting for age, sex, and the first 10 ancestry-based principal components. The genomic inflation factor for the main case–control analysis was 1.03. The genome-wide significant P value cutoff was set to $5 \times 10^{-8}$. Regional plots were created with LocusZoom[41].

**Genetic refinement and population-attributable risk**. The association between the lead SNP identified (rs6702619) and CAVS was evaluated with stratified analyses according to the presence of CAD in the case group. We also evaluated the association of this variant with the presence of CAD in the case group. The association between rs6702619 and aortic root diameter measured by ultrasound was evaluated in a subset of 900 participants (698 cases and 202 controls) for which this information was available. We performed linear regression with adjustment for age, sex, body surface area (using the DuBois and DuBois formula), and the first 10 principal components. The population-attributable risk was calculated as PAR% = $100\% \times P \times (OR - 1)/[P \times (OR - 1) + 1]$, where $P$ is the frequency of the risk allele associated with CAVS in the control group, and OR is the odds ratio calculated in the case–control cohort.

**Human aortic valve gene expression analysis**. Transcriptomic analyses were performed from 240 stenotic aortic valves collected as mentioned above. All stenotic valves were tricuspid and had a fibro-calcific remodeling score of 3 or 4[42]. Selected valves were from cases that are part of the GWAS and included 120 men and 120 women. RNA was extracted from valve leaflets, and gene expression was measured using the Illumina HumanHT-12 v4 Expression BeadChip. Standard microarray processing and QC analysis was performed[43]. The raw data were quantile normalized after log2-transformation with the lumi package in R[44]. Only one sample failed QC, leaving 239 samples for subsequent analyses. Probe sequences were mapped to RefSeq B38, GENCODE v24 B38, mRNA B38, and the human genome (GRCh38) using Bowtie, and probes not mapping to any coding region were removed. A total of 45,699 probes remained after this step. Robust fitting of linear models function (rlm) in the R statistical package MASS was used to adjust gene expression data for age and sex. Residual values were then filtered out if deviating by more than three standard deviations from the median.

**Expression quantitative trait loci**. Only subjects that passed genotyping and gene expression QCs were considered for eQTL analysis, leaving a sample size of 233. eQTLs were identified by using linear regression model and additive genotype effects as implemented in the Matrix eQTL package in R[45]. Cis-eQTLs were defined by a 2 Mb window, i.e., 1 Mb distance on either side of the SNP. eQTLs were calculated on adjusted expression traits to obtain test statistics, P values, and false discovery rate. Estimates of effect sizes were obtained with PLINK.

**Transcriptome-wide association study**. The TWAS was performed using FUSION[21]. Briefly, the valve eQTL was the reference data, including 233 individuals that passed QC for both gene expression and genotyping. This reference set was used to calculate gene expression weights using prediction models implemented in FUSION. This includes top1 (i.e., the single most significant valve eQTL-SNP as the predictor), LASSO regression, and enet (elastic net regression).

SNP data located 500 kb on both sides of the probes were used to obtain expression weights. All probes that passed QC in the valve eQTL were evaluated ($n = 45,699$). Expression weights were then combined with summary-level GWAS results to estimate association statistics between gene expression and CAVS. Genome-wide significant TWAS genes were considered at $P_{TWAS} < 0.0001$.

**Bayesian colocalization.** Summary statistics (regression coefficients and variance) from the QUEBEC-CAVS GWAS and valve eQTL results were combined using the COLOC package in R (version 2.3–6)[22]. COLOC tested for five hypotheses: H0, no eQTL and no GWAS association; H1, association with eQTL, but no GWAS; H2, association with GWAS, but no eQTL; H3, eQTL and GWAS association, but independent signals; and H4, shared eQTL and GWAS associations. The main interest is to assess whether the GWAS and eQTL signals are consistent with shared causal variants (i.e., H4). The result of this procedure is five posterior probabilities (PP0, PP1, PP2, PP3, and PP4). In practice, a high posterior probability (PP4 > 75%) indicates that the GWAS and eQTL signals colocalize.

**Mendelian randomization.** We first selected variants located within 200 kb of the gene of interest identified in the TWAS, *PALMD*, and significantly associated with *PALMD* gene expression at a threshold of $P < 0.05$.

We then performed stepwise regression (bidirectional elimination) using the *step* function in R to select SNPs independently associated with *PALMD* expression (based on the Akaike information criterion). Mendelian randomization was performed using the Wald method by regressing genetic effect estimates for CAVS risk as determined in the GWAS analysis (dependent variable) on genetic effect estimates for *PALMD* gene expression as determined in the eQTL analysis. Effect estimates were adjusted for the minor allele frequency of each variant (beta × (2 × MAF × (1 − MAF))$^{0.5}$) to better reflect the variance explained by each variant[46,47]. To determine the significance of the association, a bootstrap method was used. Predicted effects on CAVS risk and *PALMD* gene expression were sampled from a normal distribution with mean and standard deviation as determined from the GWAS and eQTL analyses. A two-tailed $P$ value was calculated using 100,000 random simulations. To determine the presence of unmeasured net pleiotropy, we performed Egger Mendelian randomization in which a non-zero *y*-intercept is allowed in order to assess violations of standard Mendelian randomization[48]. A $P$ value below 0.05 was considered as significant. A similar analysis was performed using the effect on CAVS risk as estimated in the UK Biobank. We verified the effect of the selected SNPs on the expression of the genes located nearby *PALMD*: *PLPPR4*; *FRRS1*; and *AGL* by performing linear regression analyses.

**Gene expression according to severity.** CAVS disease severity was assessed by aortic valve area, mean, and peak transvalvular gradients. The influence of normalized, age- and sex-adjusted *PALMD* expression levels on disease severity was tested in 239 cases using linear regression models. The 95% confidence intervals were estimated using the *predict* function in R.

**Replication in UK Biobank cohort.** UK Biobank is a large prospective cohort of about 500,000 individuals between 40 and 69 years old recruited from 2006 to 2010 in several centers located in the United Kingdom[49]. The present analyses were conducted under UK Biobank data application number 25205. We used genotyping data obtained from the second genetic data release, including 488,377 individuals. Samples were genotyped with the Affymetrix UK BiLEVE Axiom array or the Affymetrix UK Biobank Axiom Array. Phasing and imputation were performed centrally using a reference panel combining the Haplotype Reference Consortium (HRC) as a first choice and UK10k and 1000 Genomes Phase 3 samples for SNPs not available in HRC. Samples with call rate < 95%, outlier heterozygosity rate, gender mismatch, non-white British ancestry, related samples (second degree or closer), samples with excess third-degree relatives (> 10), or not used for relatedness calculation were excluded. Variants not on both arrays, which failed QC in more than one batch, with call rate < 95% or with minor allele frequency < 0.0001 were excluded.

CAVS diagnosis was established from hospital record, using the International Classification of Diseases version-10 (ICD10) and Office of Population Censuses and Surveys Classification of Interventions and Procedures (OPCS-4) coding. ICD10 code number I35.0 and OPCS-4 code number K26 were used. Participants with a history of rheumatic fever or rheumatic heart disease as determined by ICD10 codes I00–I02 and I05–I09 were excluded.

We performed additive logistic regression analysis using SNPTEST v2.5.2, adjusting for age, sex, and the first 10 ancestry-based principal components to evaluate the effect of the top SNP identified in our TWAS analysis. We then performed a fixed-effect meta-analysis using the inverse-variance weighted method as implemented in rmeta package version 2.16 in R. To look for other significant variants associated with CAVS in UK Biobank, we performed a GWAS from the genotyped data using additive logistic regression models in PLINK adjusting for age, sex, and the first 10 principal components. The genomic inflation factor for the main case–control analysis was 1.01. The genome-wide significant $P$ value cutoff was set to $5 \times 10^{-8}$.

Genetic associations at the *PALMD* and *LPA* loci were also performed with and without adjustment for CAD. CAD diagnoses were established from ICD10 code

numbers I20, I21, I22, I23, I24, and I25, as well as OPCS-4 code numbers K49, K502, K75, K40, K41, K45, and K46. Overall, 25,167 CAD cases were identified. In the main CAVS case–control analysis, CAD was observed in 60% of CAVS cases and 7% of controls.

**Statistical analysis.** Statistical analyses were performed with R version 3.2.3 unless otherwise specified. Two-sided $P$ values below 0.05 were considered significant unless otherwise specified.

**Western blotting.** Mineralized aortic valves were selected according to the genotype at rs6702619 (TT vs. GG). Tissues were mixed with lysis buffer (150 mM NaCl, 20 mM Tris (pH 7.5), 10% glycerol, 5 mM EGTA, 0.5 mM EDTA, 2 mM sodium vanadate, 50 mM sodium fluoride, 1% Triton X-100, 0.1% SDS, 80 mM β-glycerophosphate, 5 mM sodium pyrophosphate, 1 mM phenylmethylsulfonyl fluoride, and protease inhibitor cocktail). Mechanical lysis was performed by using a bead mill homogenizer (VWR, PA, USA), followed by centrifugation at $5000 \times g$ for 12 min at 4 °C. Supernatants were harvested and protein loading buffer (62.5 mM Tris (pH 6.8), 10% glycerol, 2% SDS, and 5% β-mercaptoethanol in $H_2O$) was added. Samples were boiled for 5 min, loaded onto polyacrylamide gels, and separated by electrophoresis, followed by transfer onto nitrocellulose membranes. Membranes were blocked with Tris-buffered saline (TBS)-Tween (5% fat-free milk), according to the manufacturer's instructions, incubated with either PALMD (Novus Biologicals, ON, Canada, #cat NBP1-88481, dilution 1:500) or GAPDH (Santa Cruz Biotechnologies, TX, USA, #cat sc-25778, dilution 1:1000) primary polyclonal antibodies overnight at 4 °C. After washing in TBS-Tween, membranes were incubated with horseradish peroxidase-labeled secondary antibodies (TransBionovo Co., Ltd, Beijing, China, #cat HS101, dilution 1:2000) and bands were detected using the clarity western ECL substrate (BioRad, ON, Canada). Images were acquired and quantified with the ChemiDocMP system (BioRad).

**Data availability.** The genome-wide association summary statistics are available in the database of Genotypes and Phenotypes (dbGaP) under accession phs001492.v1.p1. The microarray gene expression data set on human aortic valves was deposited in Gene Expression Omnibus with accession number GSE102249. All valve *cis*-eQTLs identified are listed in Supplementary Data 1.

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

## Acknowledgements

We thank the research team at the cardiac surgical database and biobank of the Institut universitaire de cardiologie et de pneumologie de Québec (IUCPQ) for their valuable assistance. This research has been conducted using the UK Biobank Resource. P.P. holds the Canada Research Chair in Valvular Heart Disease and his research program is supported by a Foundation Scheme Grant from Canadian Institutes of Health Research (Ottawa, Ontario, Canada). P.M. holds a Fonds de Recherche du Québec-Santé (FRQS) Research Chair on the Pathobiology of Calcific Aortic Valve Disease. Y.B. holds a Canada Research Chair in Genomics of Heart and Lung Diseases. This work was supported by the Heart and Stroke Foundation of Canada and the Canadian Institutes of Health Research [MOP - 102481, MOP – 137058, PJT - 153396] to Y.B.

## Author contributions

S.T., D.M.-Z., P.P., P.M. and Y.B. contributed to the conception and study design. N.G., M.R., M.-C.B., D.M.-Z., M.-A.C., R.C., F.D., P.P., P.M. and Y.B. contributed to data collection. S.T., M.L. and Y.B. contributed to data analysis. S.T., N.G., M.L., P.M. and Y.B. contributed to data interpretation.

## Additional information

**Competing interests:** The authors declare no competing interests.

