## [Peer Review File · Nature Communications]

Reviewer #1 (Remarks to the Author):

This paper by Theriault et al describes a GWAS and eQTL study of French-Canadian individuals with calcific aortic valve stenosis. The strength of this work lies in the combined use of the two techniques to identify a susceptibility locus, and a separate replication by GWAS from the UK Biobank data. The GWAS did not identify a genome wide significant result, but combined genotype-expression data from aortic valve tissue was able to identify a candidate gene in PALMD, confirmed by replicate GWAS. In addition a Mendelian randomization analysis also supported decreased expression of PALMD in CAVS. One weakness is the control group, which consists almost entirely of individuals with atherosclerotic heart disease. This is overcome by replication in the UK Biobank group. The conclusions are well supported by the data and are not overstated.

The genotyping, gene expression and statistical methods used are standard and well described. The authors applied appropriate QC metrics for genotyping and gene expression data. The controls were well matched for gender and age, and come from the same French Canadian population. A weakness is the use of individuals with atherosclerotic heart disease, given the overlap of atherosclerosis and calcific aortic valve stenosis, which may have caused a downward bias and hence the lack of any statistically significant results in the GWAS analysis. This is suggested by the Q-Q plot, showing downward tailing of the plot at high $-\log_{10}$. Comparison to a general population sample would have been preferable; however use of a French Canadian control group is important to avoid false positives due to population stratification. A comment on this limitation and balance is warranted.

Figure 2 – I would use candidate rather than causal gene.

What significance level was used for the Mendelian randomization analysis?

Which program was used for the meta-analysis (was it meta?)

Did the authors test for association with aortic measurements (aortic root, sinotubular junction, ascending aorta diameters)?

Did any of the individuals have congenital heart defects?

Reviewer #2 (Remarks to the Author):

The authors report a novel locus, PALMD, associated with risk of calcific aortic valve stenosis (CAVS), which was identified through a transcriptome-wide association scan (TWAS) analysis. Expression of PALMD was diminished in disease aortic valve tissue, and the lead eQTL SNP (*rs6702619*) was associated with CAVS risk in GWAS analysis at $p=6.12 \times 10^{-5}$). The GWAS association was reproduced in similar GWAS analysis in UK Biobank. In general, the approach employed by the authors with

respect to identifying expression signals in disease-relevant tissue and then exploring genetic associations through GWAS is a valuable one and is a rational means of assessing aetiological mechanisms and disease risk concurrently. The authors largely make appropriate inferences from their results and draw justified conclusions. I do, however, have a number of comments on the manuscript (page numbers refer to marked numbers in the manuscript).

General comments:

Selection of controls – the authors should explain why the controls in these analyses were selected from patients undergoing cardiac surgery and population-based controls with some degree of matching. The substantial difference in the prevalence of CAD between the cases and controls suggests they may differ in more respects than purely the presence or absence of CAVS.

Formal colocalisation – in order to provide a more robust link between the expression signal for PALMD and the GWAS association of the lead eQTL SNP, the authors should add formal colocalisation analysis to the manuscript. As the analyses currently stand, it is possible that the disease risk signal is not the same as the expression signal. This would add considerable weight to the manuscript.

Selection of variants for Mendelian randomisation analysis – the authors have selected SNPs for inclusion in their MR instrument based on linkage disequilibrium within a large region around the PALMD locus. This method leaves the analysis vulnerable to bias from inclusion of excess, potentially correlated variants in the instrument, and confounding through expression of others of the several genes in the region. The authors should revise their MR instrument by selecting SNPs on the basis of conditional independence and assess the effects of the SNPs in the instrument on the expression of other genes. Importantly, the use of MR-Egger does not avoid the need for such rigour in constructing instruments for MR analysis.

Specific comments:

Page 3 – “There is a long latent period...”: Latent seems a misleading adjective here. The disease is still present, but is progressive through an increasing spectrum of physiological impact before reaching the threshold for classification as ‘severe’. The authors should reconsider the word ‘latent’.

Page 3 – “...conventional cardiovascular drugs...”: ...The term ‘cardiovascular drugs’ is too vague. The authors should specify which drugs they mean by ‘cardiovascular drugs’, and this should be informed by those agents that have been tested in randomized trials for prevention or treatment of CAVS.

Page 3 – “The only treatments available...”: This should more correctly state, “the only effective treatments...”.

Page 3 – “...molecular targets to halt disease progression...”: I suggest this is revised to read, “...molecular targets to halt or slow disease progression...”.

Page 3 – “Mean age was 71.7 +/- 8.3yrs...”: The source/nature of the 8.3years figure is unclear.

Page 3 – “...the majority of controls (98%) had coronary artery disease...”: The authors should clarify here or in the supplementary material whether this definition of CAD included myocardial infarction, angiographic CAD, or a combination.

Page 16 – “Effect estimates were adjusted for the minor allele frequency of each variant”: This is an unusual step in MR analysis and should be explained and justified fully.

Reviewer #3 (Remarks to the Author):

Starting with a small discovery GWAS sample (1,009 cases/1,017 controls), complemented by eQTL mapping of 233 human aortic valve tissues, Theriault et al. have identified PALMD as a susceptibility gene for CAVS and replicated this finding using publically available UK Biobank data.

The strengths of this study include verification that the calcific aortic valve stenosis does not include individuals with bicuspid aortic valve or rheumatic heart disease.

The inclusion of transcriptome wide association analysis in a sample of 233 human aortic valve tissues is also a major strength and adds considerable validity to PALMD as a protective gene.

Does the top SNP at this locus also associate with PALMD protein expression in aortic valvular tissue?

It is unfortunate that the majority of the control group had CAD since variants causative for both CAVS and CAD may have been missed e.g. LPA in the discovery cohort - as demonstrated by the strong association with LPA in the UK Biobank sample.

Minor Comments

Table 1: Are lipid values on statin treatment? Were data available for Lp(a) levels?

Responses to referees

We would like to thank the referees for reviewing our manuscript. You will find below our responses to the comments raised by the referees. Their comments are provided verbatim in bold.

Reviewer #1:

This paper by Theriault et al describes a GWAS and eQTL study of French-Canadian individuals with calcific aortic valve stenosis. The strength of this work lies in the combined use of the two techniques to identify a susceptibility locus, and a separate replication by GWAS from the UK Biobank data. The GWAS did not identify a genome wide significant result, but combined genotype-expression data from aortic valve tissue was able to identify a candidate gene in PALMD, confirmed by replicate GWAS. In addition a Mendelian randomization analysis also supported decreased expression of PALMD in CAVS. One weakness is the control group, which consists almost entirely of individuals with atherosclerotic heart disease. This is overcome by replication in the UK Biobank group. The conclusions are well supported by the data and are not overstated.

1) The genotyping, gene expression and statistical methods used are standard and well described. The authors applied appropriate QC metrics for genotyping and gene expression data. The controls were well matched for gender and age, and come from the same French Canadian population. A weakness is the use of individuals with atherosclerotic heart disease, given the overlap of atherosclerosis and calcific aortic valve stenosis, which may have caused a downward bias and hence the lack of any statistically significant results in the GWAS analysis. This is suggested by the Q-Q plot, showing downward tailing of the plot at high $-\log_{10}$. Comparison to a general population sample would have been preferable; however use of a French Canadian control group is important to avoid false positives due to population stratification. A comment on this limitation and balance is warranted.

Thank you, this is a good point. As underlined by the reviewer, CAVS shares risk factors with coronary artery disease (CAD) and consequently it is likely that part of the genetic risk of CAVS can be explained by shared genetic variants. This is well illustrated by the LPA locus (rs10455872), which is associated with both CAD and CAVS. Moreover, a significant proportion of patients with CAVS have CAD. However, the reverse is not true as the vast majority of patients with CAD will not develop CAVS. It is likely that the CAVS risk is also driven by loci that are not shared with CAD. In our GWAS, we have selected patients with CAD without valvulopathy as a control population in order to discover non-CAD gene variants that are specific to CAVS. This strategy is a way to decrease potential associations that would be driven, at least in part, by CAD.

This strategy allowed the discovery of the *PALMD* locus, which is specifically associated with CAVS risk. The table below shows the results for the *PALMD* and *LPA* loci association with CAVS in the UK Biobank before and after adjustment for the presence of CAD.

Table R1. Genetic associations for the *PALMD* and *LPA* loci in the UK biobank before and after adjustment for the presence of CAD.

	Association with CAVS without adjustment for CAD	Association with CAVS with adjustment for CAD
PALMD (rs6702619)	OR=1.27 [1.18-1.37], p-value=6.29E-11	OR= 1.29 [1.19-1.39], p-value=1.02E-10
LPA (rs10455872)	OR= 1.56 [1.39-1.76], p-value=2.58E-14	OR= 1.39 [1.24-1.56], p-value=3.54E-08

The *PALMD* locus is not influenced by adjustment for CAD, whereas the strength of association at the *LPA* locus was lowered after adjustment, but remained significant. We have included this section in the revised manuscript (page 10):

The controls for the QUEBEC-CAVS GWAS were patients with CAD without valvulopathy. This strategy limited our ability to detect shared genetic variants with CAD. On the other hand, genes that are specific for CAVS are more likely to be found. This is illustrated by comparing the LPA locus (a known locus associated with both CAD and CAVS) and the PALMD locus in the UK biobank with and without adjustment for CAD. The association with CAVS was not influenced at the PALMD locus with an OR=1.29 ($p=1.02 \times 10^{-10}$) and OR=1.27 ($p=6.29 \times 10^{-11}$) with and without adjustment for CAD, respectively. In contrast, the strength of association at the LPA locus was lowered after adjustment for CAD (OR=1.39, $p=3.54 \times 10^{-8}$ vs OR=1.56, $p=2.58 \times 10^{-14}$).

This section was also added in the Methods (p. 18-19):

Genetic associations at the PALMD and LPA loci were also performed with and without adjustment for CAD. CAD diagnoses were established from ICD10 code numbers I20, I21, I22, I23, I24, and I25, as well as OPCS-4 code numbers K49, K502, K75, K40, K41, K45, and K46. Overall, 25,167 CAD cases were identified. In the main CAVS case-control analysis, CAD was observed in 60% of CAVS cases and 7% of controls.

2) Figure 2 – I would use candidate rather than causal gene.

The title of the figure was changed for:

Figure 2. PALMD is the candidate causal gene on the 1p21.2 CAVS susceptibility locus.

3) What significance level was used for the Mendelian randomization analysis?

We used a bootstrap method to verify the significance of the association. We performed 100,000 simulations and calculated a two-tailed p-value. We considered a p-value below 0.05 as significant since we tested only one hypothesis (association between *PALMD* expression and risk of CAVS).

We added this information in the Methods section (p. 16-17):

A P value below 0.05 was considered as significant.

4) Which program was used for the meta-analysis (was it meta?)

We modified this sentence in the Methods section (p. 18):

We then performed a fixed-effect meta-analysis using the inverse-variance weighted method as implemented in rmeta package version 2.16 in R.

5) Did the authors test for association with aortic measurements (aortic root, sinotubular junction, ascending aorta diameters)?

No, we have not tested genetic associations with aortic measurements in our original study. A GWAS on these measurements would be worthwhile, but should be the focus of a different manuscript. Data for aortic root diameter was available for 900 participants included in the study (698 cases and 202 controls). There were not enough data for sinotubular junction and ascending aorta diameters to perform analyses (total $n < 250$). We have checked whether the *PALMD* locus is associated with aortic root diameter and have included this section in the revised manuscript (p. 8):

In our study, rs6702619 was not associated with aortic root diameter ($P=0.18$), which was available in a subgroup of 900 participants (698 cases and 202 controls).

This section was also added in the Methods (p. 14):

The association between the top SNP identified (rs6702619) and aortic root diameter measured by ultrasound was evaluated in a subset of 900 participants (698 cases and 202 controls) for which this information was available. We performed linear regression with adjustment for age, sex, body surface area (using the DuBois and DuBois formula) and the first ten principal components.

6) Did any of the individuals have congenital heart defects?

Only 7 participants (3 cases and 4 controls) had interatrial communication. There was no other congenital heart defect, including bicuspid aortic valve. We included this sentence in the Methods section (p. 12):

CAVS patients and controls were free of congenital heart defects (including bicuspid aortic valve), except for 7 participants with interatrial communication (3 cases and 4 controls).

Reviewer #2:

The authors report a novel locus, PALMD, associated with risk of calcific aortic valve stenosis (CAVS), which was identified through a transcriptome-wide association scan (TWAS) analysis. Expression of PALMD was diminished in disease aortic valve tissue, and the lead eQTL SNP (rs6702619) was associated with CAVS risk in GWAS analysis at $p=6.12 \times 10^{-5}$. The GWAS association was reproduced in similar GWAS analysis in UK Biobank. In general, the approach employed by the authors with respect to identifying expression signals in disease-relevant tissue and then exploring genetic associations through GWAS is a valuable one and is a rational means of assessing aetiological mechanisms and disease risk concurrently. The authors largely make appropriate inferences from their results and draw justified conclusions. I do, however, have a number of comments on the manuscript (page numbers refer to marked numbers in the manuscript).

General comments:

1) Selection of controls – the authors should explain why the controls in these analyses were selected from patients undergoing cardiac surgery and population-based controls with some degree of matching. The substantial in the prevalence of CAD between the cases and controls suggests they may differ in more respects than purely the presence or absence of CAVS.

Cases and controls were matched for gender, age, type 2 diabetes and hypertension. BMI, cholesterol levels as well as kidney function were similar between the groups as shown in Table 1. We cannot exclude the presence of other differences between the groups, but we believe the design improves detection of genetic variants specifically associated with CAVS. Please see response to reviewer #1 (point #1).

2) Formal colocalisation – in order to provide a more robust link between the expression signal for PALMD and the GWAS association of the lead eQTL SNP, the authors should add formal colocalisation analysis to the manuscript. As the analyses currently stand, it is possible that the disease risk signal is not the same as the expression signal. This would add considerable weight to the manuscript.

We included this section in the Methods section (p. 15-16):

Bayesian colocalisation

Summary statistics, more specifically regression coefficients and their variance, from the QUEBEC-CAVS GWAS and valve eQTL results were combined using COLOC package version 2.3-6 in R. (Giambartolomei, C. et al. Bayesian test for colocalisation between pairs of genetic association studies using summary statistics. PLoS Genet, 10, e1004383 (2014)). COLOC tested for five hypotheses: H0: no eQTL and no GWAS association, H1: association with eQTL, but no GWAS, H2: association with GWAS, but no eQTL, H3: eQTL and GWAS association, but independent signals, and H4: shared eQTL and GWAS association. The main interest is to assess whether the GWAS and eQTL signals are consistent with shared causal variants (i.e. H4). The result of this procedure is five posterior probabilities (PP0, PP1, PP2, PP3 and PP4). In practice, a high posterior probability (PP4 greater than 75%) indicates that the GWAS and eQTL signals colocalize.

We also included this section in the Results section (p. 5):

In addition, formal Bayesian colocalisation (Giambartolomei, C. et al.) revealed a posterior probability of shared signals (PP4) of 0.96, which confirms that the GWAS and valve eQTL signals share the same variants at the PALMD locus.

3) Selection of variants for Mendelian randomisation analysis – the authors have selected SNPs for inclusion in their MR instrument based on linkage disequilibrium within a large region around the PALMD locus. This method leaves the analysis vulnerable to bias from inclusion of excess, potentially correlated variants in the instrument, and confounding through expression of others of the several genes in the region. The authors should revise their MR instrument by selecting SNPs on the basis of conditional independence and assess the effects of the SNPs in the instrument on the expression of other genes. Importantly, the use of MR-Egger does not avoid the need for such rigour in constructing instruments for MR analysis.

We performed additional analyses using a more rigorous instrument for MR. We narrowed the region of interest to 200kb on either side of the *PALMD* gene. We then performed stepwise regression, which led to a list of 12 SNPs independently associated with *PALMD* expression ($p < 0.05$ in univariate model and $p < 0.157$ in the model including all the SNPs). MR analyses remain significant with the use of this new instrument ($p = 0.0036$, Egger intercept $p = 0.25$).

We also tested this instrument in the UK Biobank cohort to confirm the relationship between *PALMD* gene expression and CAVS risk. The association was consistent ($p = 1.18E-05$, Egger intercept $p = 0.10$).

We verified the effect of these 12 SNPs on the expression of the genes located nearby *PALMD*: *PLPPR4*, *FRRS1* and *AGL*. Considering the number of tests performed, none of the SNPs was significantly associated with the expression of these 3 genes ($p > 0.0042$ or $0.05/12$).

We modified the following sections in the manuscript:

Methods (p. 16):

We first selected variants located within 200 kb of the gene of interest identified in the TWAS, PALMD, and significantly associated with PALMD gene expression at a threshold of $P < 0.05$. We then performed stepwise regression (bidirectional elimination) using the step function in R to select SNPs independently associated with PALMD expression (based on the Akaike information criterion).

...

A similar analysis was performed using the effect on CAVS risk as estimated in the UK Biobank.

We verified the effect of the selected SNPs on the expression of the genes located nearby *PALMD*: *PLPPR4*, *FRSS1* and *AGL* by performing linear regression analyses.

Results (p. 6):

Variants located within 200 kb of *PALMD* and associated with *PALMD* gene expression were selected using stepwise regression (**Supplementary Table 3**) to create an instrument for Mendelian randomization analyses. Effect on *PALMD* gene expression was inversely associated with the effect on CAVS risk without evidence of pleiotropy ($P=0.0036$; Egger intercept $P=0.25$; **Fig. 3a**). Considering the number of tests performed, none of the SNPs in the instrument was significantly associated with the expression of the three nearest genes (*PLPPR4*, *FRSS1* and *AGL*) ($P>0.0042$ or $0.05/12$).

And on p. 7:

Mendelian randomization using the effect on CAVS risk as estimated in the UK Biobank also pinpoints *PALMD* expression as a causal factor ($P=1.18 \times 10^{-5}$; Egger intercept $P=0.10$; **Fig. 3b**).

Figure 3. Mendelian randomization analysis of the association between *PALMD* gene expression and CAVS risk. Each circle represents one of 12 SNPs located within 200 kb of *PALMD* selected for association with *PALMD* gene expression ($P<0.05$) using stepwise regression. The blue line is the regression slope using the Wald method. The red dashed lines represent 95% confidence intervals from bootstrap. (a) Effect on CAVS risk from QUEBEC-CAVS cohort against effect on *PALMD* gene expression ($P=0.0036$). (b) Effect on CAVS risk from UK Biobank against effect on *PALMD* gene expression ($P=1.18 \times 10^{-5}$).

Specific comments:

4) Page 3 – “There is a long latent period...”: Latent seems a misleading adjective here. The disease is still present, but is progressive through an increasing spectrum of physiological impact before reaching the threshold for classification as ‘severe’. The authors should reconsider the word ‘latent’.

We have replaced by: “There is a long period of disease progression before CAVS becomes severe and symptomatic...”.

5) Page 3 – “...conventional cardiovascular drugs...”: ...The term ‘cardiovascular drugs’ is too vague. The authors should specify which drugs they mean by ‘cardiovascular drugs’, and this should be informed by those agents that have been tested in randomized trials for prevention or treatment of CAVS.

We have replaced by: “*Unfortunately, conventional cardiovascular drugs, such as statins and ACE inhibitors, are unable to stop or delay the progression of CAVS³⁻⁶.*”

The references # 3 to 6 include RCTs and reviews on the subject.

6) Page 3 – “The only treatments available...”: This should more correctly state, “the only effective treatments...”.

Thanks we have made the change.

7) Page 3 – “...molecular targets to halt disease progression...”: I suggest this is revised to read, “...molecular targets to halt or slow disease progression...”.

Thanks we have made the change.

8) Page 3 – “Mean age was 71.7 +/- 8.3yrs...”: The source/nature of the 8.3years figure is unclear.

We have replaced by: “*Mean and standard deviation for age was 71.7±8.3 years...*”.

9) Page 3 – “...the majority of controls (98%) had coronary artery disease...”: The authors should clarify here or in the supplementary material whether this definition of CAD included myocardial infarction, angiographic CAD, or a combination.

This was included in the Methods section: “*Coronary artery disease was defined as history of myocardial infarction, documented myocardial ischemia or coronary artery stenosis on coronary angiography.*”

10) Page 16 – “Effect estimates were adjusted for the minor allele frequency of each variant”: This is an unusual step in MR analysis and should be explained and justified fully.

We included this step to better reflect the variance explained by each SNP, which is dependent on its allele frequency. This concept is explained in the following article: Park et al. Nat Genet 2010, PMID 20562874. Variance explained can be considered as a good metric to estimate the effect of a SNP in MR analyses (Swerdlow et al. Int J Epidemiol 2016, PMID 27342221).

We added more details in the Methods section:

Effect estimates were adjusted for the minor allele frequency of each variant ($\beta \cdot (2 \cdot \text{MAF} \cdot (1 - \text{MAF}))^{0.5}$) to better reflect the variance explained by each variant (Park et al. Nat Genet 2010, Swerdlow et al. Int J Epidemiol 2016).

Reviewer #3:

Starting with a small discovery GWAS sample (1,009 cases/1,017 controls), complemented by eQTL mapping of 233 human aortic valve tissues, Theriault et al. have identified PALMD as a susceptibility gene for CAVS and replicated this finding using publically available UK Biobank data.

The strengths of this study include verification that the calcific aortic valve stenosis does not include individuals with bicuspid aortic valve or rheumatic heart disease.

The inclusion of transcriptome wide association analysis in a sample of 233 human aortic valve tissues is also a major strength and adds considerable validity to PALMD as a protective gene.

1) Does the top SNP at this locus also associate with PALMD protein expression in aortic valvular tissue?

This was included in the Results section (p. 5):

Concordantly, protein expression of PALMD in aortic valves was lowered in homozygotes GG compared to homozygotes TT for rs6702619 (Supplementary Fig. 4).

Supplementary Figure 4. Protein expression of PALMD by Western blot in human aortic valves. (a) Five homozygotes TT and five homozygotes GG for SNP rs6702619 were evaluated by Western blot normalized to expression of GAPDH. (b) The ratio PALMD/GAPDH is illustrated for each patient by genotyping groups. ** $p < 0.01$

This was included in the Methods section (p. 19-20):

Western blotting

Mineralized aortic valves were selected according to the genotype at rs6702619 (TT vs. GG). Tissues were mixed with lysis buffer (150mM NaCl, 20mM Tris pH7.5, 10% glycerol, 5mM EGTA, 0.5mM EDTA, 2mM sodium vanadate, 50mM sodium fluoride, 1% triton X-100, 0.1% SDS, 80mM β -glycerophosphate, 5mM sodium pyrophosphate, 1mM PMSF and protease inhibitor cocktail). Mechanical lysis was performed by using a bead mill homogenizer (VWR, PA, USA), followed by centrifugation at 5000g for 12 minutes at 4°C. Supernatants were harvested and protein loading buffer (62.5mM Tris pH6.8, 10% glycerol, 2% SDS and 5% β -mercaptoethanol in H₂O) was added. Samples were boiled 5 minutes, proteins were loaded onto polyacrylamide gels followed by electrophoresis and transferred onto nitrocellulose membranes. Membranes were blocked with TBS-tween containing 5% non-fat dry milk, according to manufacturer's instructions, incubated with either PALMD (Novus Biologicals, ON, Canada) or GAPDH (Santa Cruz Biotechnologies, TX, USA) primary antibodies

overnight at 4°C. Membranes were then washed and incubated with HRP-labeled secondary antibodies (TransBionovo Co., Ltd, Beijing, China). Detection was done using clarity western ECL substrate (BioRad, ON, Canada). Images were acquired and quantification analyses were performed using a ChemiDocMP system (BioRad, ON, Canada).

2) It is unfortunate that the majority of the control group had CAD since variants causative for both CAVS and CAD may have been missed e.g. LPA in the discovery cohort - as demonstrated by the strong association with LPA in the UK Biobank sample.

Please see response to reviewer #1 (point #1).

Minor Comments

3) Table 1: Are lipid values on statin treatment? Were data available for Lp(a) levels?

As indicated in the Methods section, cases and controls in this study are collected from patients that underwent cardiac surgery. We systematically measured standard lipid profile from fasting blood samples, but did not perform specialized measurements like Lp(a). Most of the participants were on lipid-lowering medication (73% of cases and 88% of controls). We modified this sentence in the revised manuscript (p. 12): “*In addition, fasting plasma lipids and creatinine were measured.*” and added this information in Table 1.

Reviewer #1 (Remarks to the Author):

My previous critiques have been appropriately addressed. No new critiques are noted - acceptance of the manuscript is recommended.

Kim McBride

Reviewer #2 (Remarks to the Author):

In general, the authors have addressed my previous comments adequately. The addition of the colocalisation adds support to the findings. There are two outstanding areas of concern.

I remain uncomfortable with the composition of the control and case groups with respect to the proportion of patients with CAD. I accept the authors' argument that they are seeking to identify CAVS-associated variants beyond those that share effects on CAD. However, the case and control groups as they currently stand are not comparable when assessing CAVS risk. I have two suggestions for addressing this issue: the authors should consider matching the proportion of CAD patients between the case and control groups; and, the authors should run stratified association analyses for the following groups:

[CAVS cases without CAD vs CAVS controls without CAD], [CAVS cases with CAD vs CAVS controls with CAD], [CAVS cases without CAD vs CAVS cases with CAD], [CAVS cases with CAD vs CAVS controls without CAD], and finally [all CAVS cases vs all CAVS controls]. This will allow formal comparison of the estimates for each stratum and assessment of any interaction between CAVS and CAD risk.

My second concern is around the Mendelian randomisation analysis. The modifications to the method that have been made by the authors certainly improve the approach. It would, nonetheless, be valuable to include an MR analysis using only the lead PALMD expression variant. This will inevitably have less statistical power as an instrument than the multi-variant approach but is important for excluding horizontal pleiotropy.

Reviewer #3 (Remarks to the Author):

The authors have responded in detail to my previous comments and have provided new data that considerably strengthen the manuscript. I have no further comments.

Responses to referees

We would like to thank the referees for reviewing the revised version of our manuscript. You will find below our responses to the comments raised by Reviewer #2. His/Her comments are provided verbatim in bold.

Reviewer #2:

In general, the authors have addressed my previous comments adequately. The addition of the colocalisation adds support to the findings. There are two outstanding areas of concern.

I remain uncomfortable with the composition of the control and case groups with respect to the proportion of patients with CAD. I accept the authors' argument that they are seeking to identify CAVS-associated variants beyond those that share effects on CAD. However, the case and control groups as they currently stand are not comparable when assessing CAVS risk. I have two suggestions for addressing this issue: the authors should consider matching the proportion of CAD patients between the case and control groups; and, the authors should run stratified association analyses for the following groups:

[CAVS cases without CAD vs CAVS controls without CAD], [CAVS cases with CAD vs CAVS controls with CAD], [CAVS cases without CAD vs CAVS cases with CAD], [CAVS cases with CAD vs CAVS controls without CAD], and finally [all CAVS cases vs all CAVS controls]. This will allow formal comparison of the estimates for each stratum and assessment of any interaction between CAVS and CAD risk.

In the QUEBEC-CAVS cohort, the vast majority (>95%) of participants in the control group have coronary artery disease (CAD), see Table R1 below. It is therefore not possible to match the proportion of participants with CAD in each group. It is also not possible to perform stratified analyses including a subgroup of controls without CAD since the number is too low.

Table R1. Number of CAVS cases and controls with and without CAD in the QUEBEC-CAVS cohort

	CAVS Case group	Control group
CAD	586	989
no CAD	423	28
Total	1009	1017

We performed an association analysis between our strongest expression and disease-associated SNP (rs6702619) and the presence of CAVS including only participants with CAD, i.e. CAVS cases with CAD (n=586) vs controls with CAD (n=989). Association analysis was then repeated including only CAVS cases without CAD, i.e. CAVS cases without CAD (n=423) vs controls with CAD (n=989). The results are indicated in Table R2. The results were consistent in both analyses, with an effect that seems stronger when comparing CAVS cases without CAD to controls with CAD, but the confidence intervals were overlapping.

Table R2. Analysis stratified for CAD for the association between rs6702619 and CAVS in the QUEBEC-CAVS cohort

	Beta	SE	P	OR (95% CI)
CAVS cases with CAD vs controls with CAD	0.183	0.075	0.014	1.20 (1.04 - 1.39)
CAVS cases without CAD vs controls with CAD	0.359	0.085	2.01E-05	1.43 (1.21 - 1.69)

We also performed the analysis in CAVS cases only using CAD as the outcome to verify if the variant was associated with the presence of CAD in CAVS cases. The association was not significant ($p=0.150$).

These analyses are consistent with a genetic signal that is specific for CAVS. This is further demonstrated in our manuscript using two external datasets. First, the variant is also strongly associated with CAVS in UK Biobank (a prospective population study) even after adjusting for the presence of CAD. Second, the variant is not associated with CAD in large consortia (Nikpay et al. Nat Genet 2015). Together, these results provide strong evidence that the identified locus is specifically involved in CAVS risk.

We added this section to the manuscript (page 10):

In the QUEBEC-CAVS cohort, the association of rs6702619 with CAVS stratified by the presence of CAD in cases showed consistent results with overlapping effect sizes. In addition, the association of the variant with the presence of CAD in CAVS cases was not significant ($P=0.150$).

This section was added to the Methods (page 14):

The association between the lead SNP identified (rs6702619) and CAVS was evaluated with stratified analyses according to the presence of CAD in the case group. We also evaluated the association of this variant with the presence of CAD in the case group.

My second concern is around the Mendelian randomisation analysis. The modifications to the method that have been made by the authors certainly improve the approach. It would, nonetheless, be valuable to include an MR analysis using only the lead PALMD expression variant. This will inevitably have less statistical power as an instrument than the multi-variant approach but is important for excluding horizontal pleiotropy.

We performed a Mendelian randomization analysis including only the lead variant (rs6702619), see Figure R1.

Using a bootstrap method with 100,000 simulations, the p-value was $1.37\text{E-}04$ for the direction of the association.

Figure R1. Effect of the lead variant (rs6702619) on PALMD expression in aortic valves and CAVS risk in the QUEBEC-CAVS cohort.

We added the following sentence in the Results section (page 6):

The direction of the association was the same when only the lead variant (rs6702619) was included ($P=1.37\times 10^{-4}$).

Reviewer #2 (Remarks to the Author):

The authors have responded adequately to my previous comments and have made appropriate revisions to the manuscript. I have no further comments.